# Determinants of Health-Related Quality of Life After Transarterial Chemoembolization in Hepatocellular Carcinoma Patients: A Systematic Review

**DOI:** 10.3390/jcm14113941

**Published:** 2025-06-03

**Authors:** Wei-Zheng Zhang, Jin-Qian Han, Kok-Yong Chin, Roshaya Zakaria, Nor Haty Hassan

**Affiliations:** 1Department of Nursing, Faculty of Medicine, Universiti Kebangsaan Malaysia, Kuala Lumpur 56000, Malaysia; p119154@siswa.ukm.edu.my (W.-Z.Z.); roshaya@hctm.ukm.edu.my (R.Z.); 2Department of Nursing Science, Faculty of Medicine, University of Malaya, Kuala Lumpur 50603, Malaysia; s2115921@siswa.um.edu.my; 3Department of Pharmacology, Faculty of Medicine, Universiti Kebangsaan Malaysia, Kuala Lumpur 56000, Malaysia; chinky@ukm.edu.my

**Keywords:** hepatocellular carcinoma, TACE, health-related quality of life, influencing factors, systematic review

## Abstract

**Background/Objectives:** Hepatocellular carcinoma (HCC) is a major cause of cancer-related mortality worldwide, with transarterial chemoembolization (TACE) commonly used as a palliative approach for patients who are not candidates for surgical resection. Understanding the factors that influence health-related quality of life (HRQoL) after TACE is essential for improving patient-centered care. This systematic review seeks to consolidate current evidence on the variables that impact HRQoL in HCC patients post-TACE. **Methods**: In adherence to PRISMA guidelines, a comprehensive search was conducted across five English and Chinese databases—PubMed, Scopus, Web of Science, CNKI, and Wanfang—covering studies from database inception to May 2025. Eligible studies were observational and examined factors affecting HRQoL in post-TACE HCC patients. Two independent reviewers performed screening, data extraction, and quality assessment using the Joanna Briggs Institute (JBI) Critical Appraisal Tools. **Results**: Nine studies met the inclusion criteria, including six cohort studies and three cross-sectional studies. The quality assessment rated seven studies as high quality and two as moderate. A range of validated HRQoL assessment tools was used, with the EORTC QLQ-C30 and FACT-G being the most commonly employed. The factors influencing HRQoL were grouped into five categories: (1) demographic factors (e.g., age, gender, education level); (2) clinical indicators (e.g., liver function, tumor burden); (3) psychological factors (e.g., depression, anxiety, spiritual well-being); (4) social support (e.g., financial status, coping mechanisms); and (5) physical symptoms (e.g., fatigue, pain, appetite loss). Across studies, both symptom severity and psychological distress were consistently associated with lower HRQoL. **Conclusions**: The HRQoL of HCC patients following TACE is influenced by a complex interplay of demographic, clinical, psychological, social, and symptomatic factors. Tailored, multidimensional interventions addressing these diverse aspects are crucial to optimizing recovery and improving overall well-being.

## 1. Introduction

Primary liver cancer is the sixth most frequently diagnosed malignancy globally and is the third leading cause of cancer-related mortality [1]. It encompasses three principal subtypes: hepatocellular carcinoma (HCC), intrahepatic cholangiocarcinoma (ICC), and combined hepatocellular-cholangiocarcinoma (cHCC-CCA), among which HCC accounts for approximately 75% to 85% of all cases [2]. For patients with well-preserved liver function, adequate hepatic reserve, and good overall condition, surgical resection remains the preferred therapeutic option [3]. Nevertheless, due to the absence of evident symptoms during the early stages, a significant proportion—around 80% to 85%—of HCC patients are diagnosed at an intermediate or advanced stage, by which time surgical treatment is no longer viable [4]. In such scenarios, transarterial chemoembolization (TACE) is commonly recommended as the standard of care for individuals with intermediate-stage HCC who are not suitable candidates for curative therapies [5].

Although TACE has been demonstrated to be beneficial in enhancing survival outcomes, this treatment is also associated with a variety of negative results, including pain, fatigue, nausea, and liver dysfunction, which seriously affect patients’ health-related quality of life (HRQoL) [6,7]. HRQoL is a multifaceted construct encompassing physical, psychological, and social functioning [8], which has become a vital metric in assessing the efficacy of TACE [9]. HRQoL helps researchers better understand how the condition and any related factors affect patients’ experiences [10]. Ensuring an adequate HRQoL is an expectation for HCC patients undergoing TACE treatment [11]. However, evidence suggests that the overall HRQoL in HCC patients post-TACE remained inadequate, especially during the early recovery phase [12]. With the growing emphasis on patient-centered care in oncology nursing [13], understanding the influencing factors on HRQoL in HCC patients post-TACE is essential for developing targeted interventions to improve their well-being and clinical outcomes.

Recently, numerous studies have investigated the influencing factors on HRQoL in HCC patients post-TACE. Severe postembolization syndrome (PSE) can lengthen hospital stays, add to the patients’ physical and mental suffering, and eventually lower patients’ HRQoL [7]. In a network analysis, symptomatic distress, anxiety, and depression in HCC patients post-TACE were measured by the researchers, finding that changes in HRQoL were associated with symptomatic distress, anxiety, and depression [14]. In addition, an integrated analysis examined symptom clusters and symptom interference in HCC patients following TACE, revealing that exhaustion was the most severe symptom, succeeded by sleep disturbance, distress, depression, and loss of appetite, which all have negatively impacted patients’ HRQoL [15]. Although several studies have identified potential influencing factors on HRQoL in HCC patients post-TACE, the findings remain disjointed and occasionally contradictory. Furthermore, there is presently no thorough synthesis available to properly assess and summarize these influencing factors. This gap restricts medical staff’s capacity to devise focused interventions to enhance patients’ overall well-being.

Therefore, the aim of this systematic review is to identify, synthesize, and critically evaluate the existing information regarding factors that influence HRQoL in HCC patients post-TACE. This review aims to establish a theoretical basis for future clinical decision-making, nursing care, and the formulation of patient-centered interventions by elucidating these elements.

## 2. Materials and Methods

This systematic review followed the guidelines outlined in the Preferred Reporting Items for Systematic Reviews and Meta-Analyses (PRISMA) checklist [16] (see Appendix A). The review protocol was prospectively registered with the International Prospective Register of Systematic Reviews (PROSPERO) under the registration number CRD420250656344.

### 2.1. Search Strategy

A systematic and comprehensive literature search was carried out across five electronic databases: PubMed, Scopus, Web of Science, Wanfang, and CNKI, from database inception to May 2025. The objective was to identify studies exploring factors associated with HRQoL among patients with HCC following TACE. As no prior systematic review addressing this specific topic was found, no restrictions on publication year were imposed on the search. Only studies published in English or Chinese were considered eligible. The search strategy incorporated a combination of controlled vocabulary and free-text terms. In PubMed, for instance, the following search terms were used: (“factor*” OR “influenc*” OR “determinant*” OR “predictor*” OR “correlate*” OR “contributor*”) AND (“qol” OR “HRQoL” OR “quality of life”) AND (“TACE” OR “transarterial chemoembolization” OR “chemoembolization”). For Chinese databases such as CNKI, the key terms included the following: (“生活质量” (quality of life)) AND (“TACE”OR “灌注术” (Transcatheter Arterial Chemoembolization) OR “栓塞术” (Hepatic artery embolization)) AND (“影响因素” (influencing factors)). A complete list of search strategies for all databases is provided in Appendix A.

### 2.2. Inclusion and Exclusion Criteria

Studies were deemed eligible for inclusion if they satisfied the following conditions: (1) participants had a confirmed diagnosis of HCC; (2) TACE was the sole therapeutic intervention administered; (3) the study explored variables influencing HRQoL following TACE; (4) the research design was observational in nature, including cohort, case–control, or cross-sectional studies; (5) HRQoL was assessed using validated measurement instruments; and (6) the publication language was either English or Chinese.

Our review was limited to observational studies, as these designs are most effective for identifying associations between patient characteristics or treatment-related factors and HRQoL outcomes. Qualitative or mixed-methods studies, although useful for examining patient experiences, generally do not facilitate the quantification of associations, which was the main objective of this review.

Studies were excluded if they met any of the following criteria: (1) the study population included patients with metastatic HCC; (2) the publication type was a case report, review, abstract, study protocol, registry entry, conference proceeding, or commentary; (3) essential data for analysis were unavailable; or (4) the full text could not be retrieved despite contacting the authors or utilizing institutional access tools.

### 2.3. Study Selection

All identified records were first imported into EndNote Version 20 (Clarivate, PA, USA) for initial management, where automatic deduplication was performed. Subsequently, two independent reviewers (WZZ and JQH) screened the titles and abstracts based on the predefined inclusion and exclusion criteria, eliminating studies deemed irrelevant. For those articles meeting the preliminary eligibility requirements, full texts were retrieved and examined in detail by the same reviewers. Any disagreements during the selection process were resolved through discussion with a third reviewer (KYC). A visual summary of the screening and selection process is provided in Figure 1.

### 2.4. Data Extraction

Two reviewers (WZZ and JQH) independently extracted relevant information using a standardized Excel template. Any disagreements were resolved through consultation with a third reviewer (KYC). The extracted data included the following details: (1) first author, country of origin, and publication year; (2) type of study design; (3) total number of participants; (4) instrument used to measure HRQoL; (5) timing of HRQoL assessment; (6) primary factors influencing HRQoL; and (7) main findings or conclusions.

### 2.5. Data Synthesis

A narrative synthesis was performed to categorize and summarize the factors influencing HRQoL as identified in the selected studies. Following data extraction, these factors were organized into five distinct groups based on their content and features, to aid in comparison and analysis. The five categories were demographic, clinical, psychological, social support, and physical factors. Within each category, both similarities and variations across studies were examined. Given the variation in study methodologies, participant populations, and assessment instruments, a meta-analysis was deemed unsuitable.

### 2.6. Quality Assessment

Two reviewers (WZZ and JQH) independently assessed the methodological quality of the studies included in the review, employing the relevant tools from the Joanna Briggs Institute (JBI) Critical Appraisal Tools, with the choice of tool based on the study design. For cross-sectional studies, the JBI Checklist for Analytical Cross-sectional Studies (8 items) was used [17], while the JBI Checklist for Cohort Studies (12 items) was applied to cohort studies [18]. Each study was evaluated against these criteria with four possible responses: “Yes”, “No”, “Unclear”, and “Not applicable”. Based on the percentage of “Yes” responses, studies were classified as high (≥70%), moderate (50–69%), or low quality (<50%). Overall, the studies included in the review displayed acceptable methodological rigor, and no substantial bias risks were identified [19].

## 3. Results

### 3.1. Characteristics of Studies

As shown in Figure 1, the initial search across databases resulted in 642 records. After duplicates were removed, 487 distinct articles remained. A subsequent review of titles and abstracts identified 18 studies as potentially relevant. Following a thorough evaluation of the full texts, based on the established inclusion and exclusion criteria, nine studies ultimately met the eligibility requirements and were included in this systematic review.

The characteristics of the included studies showed notable variation across several dimensions, including geographical location, publication year, research design, sample size, and follow-up duration. Specifically, the studies were carried out in Germany (*n* = 2), Taiwan and China (*n* = 2), mainland China (*n* = 4), and Canada (*n* = 1). Of these, six employed a cohort design, while three were cross-sectional studies. Participant numbers ranged from 48 to 348, and follow-up intervals spanned from the third postoperative day to as long as 12 months, contingent upon each study’s specific aims. A comprehensive summary is presented in Table 1.

### 3.2. Quality Assessment of the Included Studies

The quality of the included studies is summarized in Table 2 and Table 3. Six cohort studies were assessed using the JBI Critical Appraisal Checklist for Cohort Studies (The complete description of the criteria is provided in Appendix A), with five rated as high quality (75–80%) and one as moderate quality. Common limitations among cohort studies included inconsistent exposure measurement between groups (item 3, rated “No” in all studies) and insufficient reporting on follow-up duration and completeness (items 9–11), which may affect internal validity. Differential measurement methods between exposed and unexposed groups may introduce selection bias, and the lack of follow-up strategies raises concerns about attrition bias, particularly in relation to longitudinal outcomes. Although confounding was adequately addressed, these issues related to exposure assessment and follow-up procedures suggest that effect estimates should be interpreted with caution. Three cross-sectional studies were evaluated using the JBI Checklist for Analytical Cross-Sectional Studies (the complete description of the criteria is provided in Appendix A), all of which were rated as high quality. However, confounding factors were often not clearly identified (item 5, rated “No” in two studies) and insufficient strategies to address them (item 6, rated “No” in all three studies), which may introduce bias. A subset of studies failed to identify potential confounding factors or implement adjustment strategies, while others acknowledged confounders but lacked explicit methods to address them. These gaps may lead to residual confounding bias, potentially inflating or obscuring the observed exposure-outcome associations. Despite these limitations, all studies demonstrated rigorous measurement of exposures and outcomes using validated criteria, alongside appropriate statistical analyses, supporting the robustness of core methodological aspects. Overall, seven studies were rated as high quality and two as moderate, providing a generally reliable basis for synthesis.

### 3.3. HRQoL Assessment Tool and Time Point

A variety of validated assessment tools were utilized across the included investigations to evaluate HRQoL in HCC patients receiving TACE. The most commonly used tools were the EORTC QLQ-C30, either alone or in combination with the EORTC QLQ-HCC18, used in three studies [20,21,24]. Other questionnaires comprised the Functional Assessment of Cancer Therapy–General (FACT-G) [11,27], the Short Form-12 (SF-12) [22], the World Health Organization Quality of Life-BREF (WHOQOL-BREF) [23], the Quality of Life Instrument for Liver Cancer (QoL-LC V2.0) [25,26], and the Functional Assessment of Chronic Illness Therapy-Spiritual Well-being (FACIT-Sp12) [26].

Assessment time points varied among studies. Most studies conducted HRQoL assessments both before and after TACE, with follow-up durations ranging from postoperative day 3 [26,27] to 12 months post-TACE [23]. Some studies included multiple follow-up points, such as at 2 weeks [20,21], 6 weeks [25], or 3 months after discharge [24], while others assessed HRQoL cross-sectionally at discharge [11] or over several intervals post-discharge [22]. The specific HRQoL domains reported across studies included global health status, physical functioning, psychological functioning, social functioning, and symptom burden. Detailed information on the HRQoL instruments, time points, and domains assessed is provided in Table 4.

### 3.4. Classification and Characteristics of HRQoL Assessment Tools

Table 5 presents detailed characteristics of each HRQoL instrument, including scoring range, interpretation, and domains assessed. The included studies utilized seven different HRQoL assessment tools, each varying in structure, scoring, and domain focus. The most frequently employed instruments were the EORTC QLQ-C30 (*n* = 3) and the FACT-G (*n* = 2), both of which are cancer-specific tools with well-established psychometric properties. The EORTC QLQ-C30, along with its liver cancer-specific module EORTC QLQ-HCC18, evaluates various functional aspects, including physical, role, emotional, cognitive, and social functioning, in addition to symptom burden such as pain, fatigue, and fever. Each domain is scored on a scale from 0 to 100; higher functional scores represent better HRQoL, whereas higher symptom scores indicate more severe symptoms.

The FACT-G measures general cancer-related HRQoL in four areas: physical, social/family, emotional, and functional well-being. It provides a total score ranging from 0 to 108, with higher scores indicating better HRQoL. The liver cancer-specific QoL-LC V2.0 was utilized in two studies and assesses physical, psychological, and social functioning, along with both general and liver cancer-related symptoms. Each question is rated using a 5-point Likert scale and then transformed into a score from 0 to 100. Other assessment tools include the SF-12, which evaluates both physical and mental aspects of general health; the WHOQOL-BREF, which covers physical, psychological, social, and environmental well-being; and the FACIT-Sp12, a unique tool focusing on spiritual health. For all these instruments, higher scores correlate with better HRQoL in their respective domains.

### 3.5. Influencing Factors on HRQoL of Studies Included

Table 6 shows that the included studies identified a wide range of factors that influence HRQoL in patients with HCC following TACE. These influencing factors can be grouped into symptoms, psychological status, demographic variables, and clinical parameters.

Hartrumpf et al. [20] reported that symptom distress, including fatigue and pain, along with psychological factors like anxiety and depression, significantly detrimentally affected physical HRQoL. Demographic factors such as education level, marital status, and employment status also influenced HRQoL outcomes. Similarly, Hinrichs et al. [21] emphasized the importance of fatigue, pain, and emotional distress, noting that HRQoL, particularly physical functioning, improved over time but remained influenced by depression and anxiety. Chen et al. [11] found that both physical and mental dimensions of HRQoL were significantly influenced by age, gender, disease status, as well as levels of depression and symptom distress. Shun et al. [22] examined clinical indicators such as tumor response, AFP levels, and MELD scores, observing that although HRQoL remained stable during the first year, a decline in physical health was associated with tumor progression.

Eltawil et al. [23] suggested that lack of disease awareness, prior liver cancer surgery, and lower education levels were linked to poorer HRQoL and increased psychological distress. In a more recent study, Zhao et al. [24] identified gender, living location, income, pain, and spiritual well-being as significant predictors of HRQoL, highlighting the importance of both psychosocial and clinical dimensions. Zheng et al. [25] underscored the role of older age, poor liver function (Child–Pugh classification), TNM stage, and liver pain as negative predictors of HRQoL. Liu et al. [26] also emphasized gender, residence, family income, pain, and spiritual health as key influencing factors. Lastly, Cao et al. [27] found that pain, fatigue, appetite loss, sadness, and abdominal bloating were the most significant symptoms affecting HRQoL after TACE.

### 3.6. Categorization of Influencing Factors on HRQoL

To further clarify the determinants of HRQoL, influencing factors identified in the included studies were categorized into five overarching domains: demographic, clinical, psychological, social support, and physical factors. The detailed information is illustrated in Table 7. Demographic variables such as age, gender, and education level were found to influence HRQoL through associations with health literacy, care access, and coping capacity [11,22,24,25,26]. Clinical factors, including tumor characteristics and liver function scores, were linked to disease severity and treatment outcome [11,20,24,27]. Psychological influences, such as anxiety, depression, and spiritual health, were shown to mediate the emotional and cognitive dimensions of HRQoL [20,22,24,27]. Social support factors, including income and coping mechanisms, played a buffering role against distress [24,25]. Physical symptoms like pain and appetite loss were central in determining physical well-being [11,27].

## 4. Discussion

This systematic review summarizes influencing factors among patients with HCC undergoing TACE from nine studies. These factors were categorized into five domains, including demographic factors, clinical factors, psychological factors, social support factors, and physical factors. These findings emphasize the multifaceted character of HRQoL and illustrate the complex interactions of multiple variables influencing the patients’ outcomes following TACE.

### 4.1. Demographic Factors

Demographic variables were consistently identified as significant determinants of HRQoL among HCC patients post-TACE [28]. Age was frequently associated with physical functioning, with older patients reporting worse HRQoL outcomes due to comorbidities and diminished physiological reserve [29]. In addition, the literature has indicated that female patients tend to report lower HRQoL compared to males [30], mainly attributable to higher psychological susceptibility and symptom awareness [31,32]. Also, advanced educational attainment relates to improved HRQoL, likely due to those with higher education being more adept at comprehending disease-related information, engaging in treatment decisions, and implementing effective self-management strategies [33]. Marital status and family support were shown to provide emotional stability and instrumental assistance, buffering the negative impact of disease burden [34]. Employment status was another crucial factor, as unemployed patients often experienced greater psychological stress and financial strain, contributing to reduced overall well-being [35]. Furthermore, patients living in rural areas or under-resourced regions were more likely to report lower HRQoL compared to their urban counterparts [22,24]. These findings highlight the importance of considering demographic diversity in the design of patient-centered interventions and follow-up care strategies.

### 4.2. Clinical Factors

Clinical factors such as TACE treatment, tumor type and size, TNM stage, Child-Pugh classification, and history of hepatitis or liver resection were identified as important determinants of HRQoL in HCC patients post-TACE [11,20,24,27]. There is evidence that patients with advanced TNM stage or larger tumor burden tend to report lower HRQoL due to more severe physical symptoms and higher treatment toxicity, particularly affecting physical and functional well-being [36,37]. Similarly, impaired liver function, reflected by a higher Child–Pugh class, tends to limit physiological reserves and increases vulnerability to complications [38]. In addition, repeated TACE treatments may accumulate hepatic injury and prolong recovery time, which can negatively influence perceived HRQoL, as supported by prior findings [39]. Furthermore, a history of hepatitis or previous liver resection may influence patients’ baseline liver status and response to treatment, thereby impacting long-term HRQoL outcomes, which is consistent with earlier studies emphasizing the history of liver disease in shaping patients’ health status [40].

Furthermore, the decline in liver function following TACE is a clinically relevant issue that may directly undermine HRQoL [41]. Numerous studies have indicated that TACE may result in hepatic decompensation, particularly in those with marginal liver function [42,43,44]. This is frequently indicated by deteriorating Child–Pugh scores following the surgery, which are significantly correlated with heightened morbidity and diminished quality of life. To avoid this risk, current guidelines recommend employing selective or superselective transarterial procedures during TACE, which can reduce damage to non-tumorous liver tissue and maintain hepatic function [45,46]. Nevertheless, the majority of the studies included failed to provide specifics regarding the TACE approach employed, therefore constraining the assessment of its effect on liver function and HRQoL. Future research should incorporate the reporting of procedural variables to more accurately evaluate their impact on outcomes.

### 4.3. Psychological Factors

The impact of psychological factors on HRQoL was evident across several studies [20,22,24,27]. Anxiety and depression were among the most commonly reported conditions, often leading to diminished emotional well-being and decreased treatment compliance. Prior research has also indicated that patients experiencing these symptoms are more likely to report lower overall HRQoL [47]. Significantly, spiritual health seems to fulfill a protective role. Patients exhibiting a heightened sense of spiritual well-being generally reported superior emotional coping and enhanced HRQoL, supporting past research findings [48], which indicates that spirituality may play a beneficial role in improving HRQoL through mechanisms such as emotional regulation, meaning-making, and adaptive coping. Hence, incorporating spiritual support into future patient-centered interventions may be considered to improve overall well-being in cancer care for HCC patients following TACE.

### 4.4. Social Support Factors

Social and familial support significantly influences the HRQoL of HCC patients following TACE [24,25]. Adequate family income has been identified as a protective factor, likely because it reduces financial stress related to treatment costs and facilitates access to medical resources and supportive services. In several studies, patients with stable economic conditions reported better emotional and functional well-being [33,49]. Moreover, strong family support was consistently linked to improved coping capacity, lower psychological distress, and enhanced overall HRQoL [50]. A prior study reported that social support from peers, healthcare providers, or broader community networks also contributed positively by alleviating feelings of isolation and fostering a sense of belonging [51]. Therefore, strengthening support systems both within and outside the family may serve as an effective strategy to improve HCC patients post-TACE.

### 4.5. Physical Factors

Physical factors such as fatigue, abdominal distension, pain, and appetite loss were commonly reported and had a substantial impact on HCC patients post-TACE [11,27]. Fatigue was the most frequently mentioned symptom, which may result from a combination of tumor necrosis-induced inflammatory responses, temporary hepatic dysfunction, and associated symptoms such as pain, insomnia, and abdominal discomfort. More than half of the patients experienced moderate to severe abdominal pain post-TACE [52], which highlights the need for proactive pain management strategies to improve HRQoL in HCC patients post-TACE. In addition to its impact on HRQoL, abdominal pain and fever are often indicative of post-embolization syndrome, a common complication following TACE [53,54]. This syndrome is generally associated with transient hypertransaminasemia resulting from hepatic ischemia and inflammation. Recent evidence suggests that transient elevations in liver enzymes following TACE may positively correlate with objective radiological tumor response, potentially serving as a prognostic indicator for personalized treatment planning [55]. This underscores the dual function of post-embolization symptoms, serving both as contributors to symptom burden and as potential indicators of therapeutic efficacy. Building on our previous scoping review [56], which synthesized evidence of pain management on both pharmacological and non-pharmacological strategies, we highlight the lack of standardized pain management protocols in TACE. Therefore, targeted interventions addressing these physical symptoms are crucial for enhancing recovery and improving overall HRQoL in HCC patients post-TACE.

### 4.6. Measurement Heterogeneity

This review’s synthesis of findings is influenced by the significant variability in HRQoL measurement tools and follow-up periods across the studies examined. This variation may influence the internal coherence and interpretability of the aggregated results. Some tools, such as the EORTC QLQ-C30, are designed specifically for cancer and encompass a wide array of symptoms and functions [57]. In contrast, tools like the SF-12 are more general and may lack sensitivity to specific disease or treatment effects [58]. HRQoL evaluated shortly after TACE may indicate immediate procedural side effects, while evaluations at 6 or 12 months may represent recovery, adaptation, or disease progression [59]. The synthesized conclusions should be regarded as indicative patterns rather than definitive quantifications. The domain-based categorization facilitates structured interpretation; however, it is crucial to acknowledge that the observed associations may vary in strength or direction based on the measurement context. The methodological differences highlight the necessity of cautious interpretation of the synthesized findings, as they may restrict the overall validity and generalizability of the conclusions.

### 4.7. Potential Impact of Evolving TACE Techniques on HRQoL Outcomes

The studies reviewed cover a period from 2012 to 2024, highlighting significant advancements in TACE techniques. These advancements include the transition from conventional lipiodol-based TACE to drug-eluting bead TACE (DEB-TACE), the implementation of image-guided delivery methods such as cone-beam CT, and the integration of combination therapies with systemic agents [60]. Recent methodologies have demonstrated a correlation with decreased systemic toxicity, less severe post-embolization symptoms, and expedited recovery [61,62,63], factors that can enhance HRQoL outcomes. Consequently, discrepancies in HRQoL findings among studies may be attributed in part to these technological advancements rather than actual variations in patient- or treatment-related factors. The temporal evolution highlights the necessity of cautious interpretation of pooled findings and indicates that future syntheses could improve by stratifying results according to TACE modality or time period.

### 4.8. Implications for Clinical Practice

This review’s findings emphasize the multidimensional character of HRQoL in HCC patients post-TACE, emphasizing the necessity for comprehensive and personalized care solutions. Firstly, clinicians must consistently evaluate both physical symptoms, such as pain and exhaustion, and psychological discomfort, including anxiety and depression, as these factors profoundly influence patients’ overall well-being. Interventions designed to improve patients’ psychological resilience and spiritual well-being may be especially beneficial. A recent randomized controlled trial conducted in China revealed that a structured spiritual care plan markedly improved spiritual health, decreased anxiety and depression, and enhanced quality of life among advanced cancer patients, thereby supporting the feasibility and effectiveness of such interventions in standard oncology practices [64]. Moreover, a comprehensive cross-sectional study indicated a significant association between spiritual well-being and improved quality of life, as well as reduced levels of anxiety and depression in cancer patients. The authors proposed that spiritual support should extend beyond religious individuals and advocated for its integration with psychological counseling to more effectively meet patients’ emotional and existential needs [65]. Palliative care for post-TACE patients with HCC generally encompasses symptom management, such as pain, fatigue, nausea, and coordination by a multidisciplinary team [66]. A systematic review reported that comprehensive palliative care interventions are effective in enhancing overall quality of life and alleviating symptom burden among patients [67]. Additionally, demographic and socioeconomic variables, including age, educational attainment, familial support, and financial condition, must be considered in treatment planning and patient education. Furthermore, enhancing social support among patients, families, and healthcare providers can alleviate psychological distress and foster improved health outcomes. In addition, customized health education and self-management assistance for patients with lower educational attainment or insufficient health literacy may enhance HRQoL in HCC patients post-TACE. Notably, a multidisciplinary team approach that incorporates psychosocial support, nutritional counseling, and spiritual care is strongly recommended to meet the varied needs of this patient population.

### 4.9. Limitations

This systematic review has several limitations that should be acknowledged. Firstly, despite conducting a comprehensive search of both Chinese and English databases, only nine studies met the inclusion criteria, which may restrict the scope of the findings. Secondly, all included studies were observational in design, consisting of six cohort studies and three cross-sectional studies, which limits the ability to draw definitive conclusions due to the potential influence of confounding variables and biases. Thirdly, while seven studies were rated as high quality and two as moderate quality, the diversity in study designs and assessment tools may have introduced bias and affected the consistency of the results. Finally, due to the heterogeneity in patient populations, HRQoL measurement instruments, and follow-up timing after TACE, a quantitative meta-analysis was not feasible. Consequently, only a qualitative synthesis was performed, which may weaken the statistical power of the conclusions. These limitations should be taken into account when interpreting the findings, and future research should focus on using standardized, high-quality methodologies to better understand the factors influencing HRQoL in HCC patients post-TACE.

## 5. Conclusions

This systematic review synthesized current evidence on factors influencing HRQoL in HCC patients post-TACE. The findings indicate that HRQoL is shaped by five major categories of factors: demographic (e.g., age, gender, education level), clinical (e.g., liver function, tumor size), psychological (e.g., anxiety, depression), social support (e.g., family or caregiver support), and physical factors (e.g., pain, fatigue). These results highlight the need for comprehensive and individualized supportive care strategies to improve patient outcomes.

## Figures and Tables

**Figure 1 jcm-14-03941-f001:**
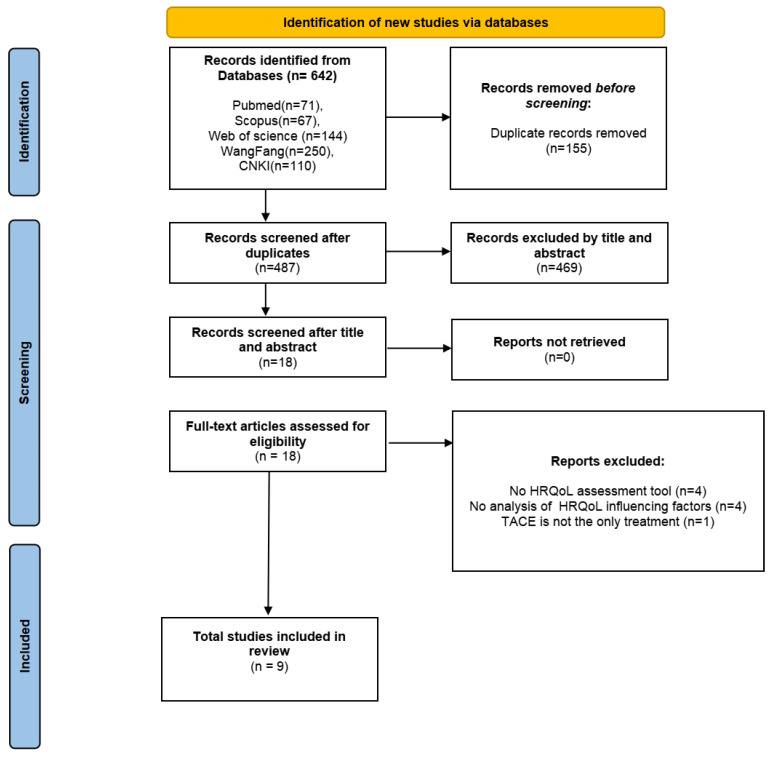
Flow diagram outlining the study identification and selection process.

**Table 1 jcm-14-03941-t001:** Characteristics of included studies.

Author	Country	Year	Study Design	Sample Size	Follow-Up Duration
Hartrumpf et al. [20]	Germany	2018	Cohort	148	2 weeks
Hinrichs et al. [21]	Germany	2017	Cohort	79	2 weeks
Chen et al. [11]	Taiwan, China	2022	Cross-sectional	98	2 months
Shun et al. [22]	Taiwan, China	2012	Cohort	89	2 months
Eltawil et al. [23]	Canada	2012	Cohort	48	12 months
Zhao et al. [24]	China	2024	Cohort	348	3 months
Zheng et al. [25]	China	2020	Cohort	130	6 weeks
Liu et al. [26]	China	2020	Cross-sectional	161	postoperative day 3
Cao et al. [27]	China	2012	Cross-sectional	142	postoperative day 3

**Table 2 jcm-14-03941-t002:** Quality assessment of the studies based on the JBI Critical Appraisal Checklist for Cohort Studies.

Author	1	2	3	4	5	6	7	8	9	10	11	12	% Yes	Quality Level	Include	Key Methodological Weaknesses
Hartrumpf et al. [20]	Y	Y	N	Y	Y	Y	Y	Y	U	U	U	Y	66.7%	M	✔	Exposure measurement inconsistency between groups Incomplete follow-up reporting
Hinrichs et al. [21]	Y	Y	N	Y	Y	Y	Y	Y	U	U	U	Y	80%	M	✔	same as above
Shun et al. [22]	Y	Y	N	Y	Y	Y	Y	Y	Y	U	U	Y	75%	H	✔	same as above
Eltawil et al. [23]	Y	Y	N	Y	Y	Y	Y	Y	Y	U	U	Y	75%	H	✔	same as above
Zhao et al. [24]	Y	Y	N	Y	Y	Y	Y	Y	Y	U	U	Y	75%	H	✔	same as above
Zheng et al. [25]	Y	Y	N	Y	Y	Y	Y	Y	Y	U	U	Y	75%	H	✔	same as above

Abbreviations: Y: Yes; N: No; U: Unclear; H: High; M: Moderate. The meaning of the numbers used in this table can be found in Appendix A.

**Table 3 jcm-14-03941-t003:** Quality assessment of studies based on the JBI Critical Appraisal Checklist for Analytical Cross-sectional Studies.

Author and Year	1	2	3	4	5	6	7	8	% Yes	Quality Level	Included	Key Methodological Weaknesses
Chen et al. [11]	Y	Y	Y	Y	N	N	Y	Y	75%	H	✔	Confounding factors not identified No strategies to address confounding
Liu et al. [26]	Y	Y	Y	Y	Y	N	Y	Y	87.5%	H	✔	No strategies to address confounding
Cao et al. [27]	Y	Y	Y	Y	N	N	Y	Y	75%	H	✔	Confounding factors not identified No strategies to address confounding

Abbreviations: Y: Yes; N: No; H: High. The meaning of the numbers used in this table can be found in Appendix A.

**Table 4 jcm-14-03941-t004:** HRQoL assessment tool.

Author	Country	Year	HRQoL Tool	Assessment Time Point	Specific Domains Reported
Hartrumpf et al. [20]	Germany	2018	EORTC QLQ-C30 EORTC QLQ-HCC18	Pre-TACE 14 days post-TACE	Global health status Physical functioning Symptom scales: pain, nausea and vomiting, fever
Hinrichs et al. [21]	Germany	2017	EORTC QLQ-C30EORTC QLQ-HCC18	Pre-TACE 2 weeks post-TACE	Global health score Physical functioning Role functioning Social functioning Symptoms: fatigue, loss of appetite, pain, nausea and vomiting, abdominal swelling
Chen et al. [11]	Taiwan, China	2022	FACT-G	Discharge day post-TACE	Physical well-being Social and family well-being Emotional well-being Functional well-being
Shun et al. [22]	Taiwan, China	2012	SF-12	Within 3 days prior to discharge 4 weeks after discharge 8 weeks after discharge	PCS: Physical functioning Role physical Bodily pain General health MCS: Vitality Social functioning Role emotional Mental health
Eltawil et al. [23]	Canada	2012	WHOQOL-BREF	Pre-TACE Every 3 months post-TACE over a 12-month follow-up	Physical health Psychological health Social relationship Environmental health
Zhao et al. [24]	China	2024	EORTC QLQ-C30	Pre-TACE 3 months post-discharge	Global health status Physical functioning Symptom scales: pain, nausea and vomiting, fever
Zheng et al. [25]	China	2020	QoL-LC V2.0	Pre-TACE 6 weeks post-TACE	Physical functioning Psychological functioning Symptoms and impact Social functioning
Liu et al. [26]	China	2020	QOL-LC V2.0 FACIT-Sp12	3 days post-TACE	Physical functioning Psychological functioning Symptoms Social functioning
Cao et al. [27]	China	2012	FACT-G	3 days post-TACE	Physical well-being Social and family well-being Emotional well-being Functional well-being

**Table 5 jcm-14-03941-t005:** Classification and characteristics of HRQoL assessment tools of included studies.

Tool	Score Range	Scoring Interpretation	Number of Items	Domain	Author and Year
EORTC QLQ-C30 (*n* = 3 studies)	0–100 per domain	Higher functional scores = better HRQoL; higher symptom scores = worse symptoms.	30	Physical Role Emotional Cognitive Social	Hartrumpf et al., 2018 [20]; Hinrichs et al., 2017 [21]; Zhao et al., 2024 [24]
EORTC QLQ-HCC18 (*n* = 2 studies)	0–100 per domain	Higher functional scores = better HRQoL; higher symptom scores = worse symptoms.	18	Fatigue Body image Jaundice Nutrition Pain Fever Abdominal Swelling	Hartrumpf et al., 2018 [20]; Hinrichs et al., 2017 [21]
FACT-G (*n* = 2 studies)	0–108	The final score is calculated by adding all subscale scores. HRQoL improves with higher overall scores.	27	Physical well-being Social and family well-being Emotional well-being Functional well-being	Chen et al., 2022 [11]; Cao et al., 2012 [27]
QoL-LC V2.0 (*n* = 2 studies)	0–100 per domain	More points mean greater HRQoL. Each item was assessed on a 5-point Likert scale (1 = not at all to 5 = very lot) and then converted to a 0–100 scale.	35	Physical function Psychological function social function Common symptoms and side Effects Specific Symptoms of liver cancer	Zheng et al., 2020 [25]; Liu et al., 2020 [26]
SF-12 (*n* = 1 study)	0–100	More points mean better health. The norm-based PCS and MCS scores are standardized to the general population. The mean score is 50, while the standard deviation is 10.	12	PCS: Physical functioning Role physical Bodily pain General health MCS: Vitality Social functioning Role emotional Mental health	Shun et al., 2012 [22]
WHOQOL-BREF (*n* = 1 study)	4–20 for each domain; 0–100 for overall HRQoL	HRQoL improves with higher scores. Higher scores indicate better health, functioning, or well-being in each domain. The 4–20 range is converted to a 0–100 scale to calculate HRQoL.	26	Physical health Psychological health Social relationshipEnvironmental health	Eltawil et al., 2012 [23]
FACIT-Sp12 (*n* = 1 study)	0–48	Higher scores indicate better levels of spiritual well-being.	12	Spiritual Well-Being	Liu et al., 2020 [26]

**Table 6 jcm-14-03941-t006:** Factors influencing HRQoL in the studies included.

Author and Year	Positive Predictors	Negative Predictors
Hartrumpf et al., 2018 [20]	Education level, marital status, employment status	Fatigue, pain, anxiety, depression, symptom distress
Hinrichs et al., 2017 [21]	Physical functioning, role functioning	Fatigue, pain, anxiety, depression
Chen et al., 2022 [11]	NA	Age, gender, depression, symptom distress, Disease status
Shun et al., 2012 [22]	Stable AFP levels, MELD score	Tumor size
Eltawil et al., 2012 [23]	Disease awareness	Age, previous liver cancer surgery
Zhao et al., 2024 [24]	Spiritual health	Gender, living location, family income, pain, Hepatitis B history, family cancer history
Zheng et al., 2020 [25]	Family support	Older age, Child–Pugh classification (worse liver function), liver pain, TNM stage
Liu et al., 2020 [26]	Spiritual health	Gender, residence, family income, hepatitis history, family cancer history, pain
Cao et al., 2012 [27]	NA	Pain, sadness, appetite loss, abdominal bloating, fatigue

Abbreviations: AFP: Alpha fetoprotein; MELD: Model for end-stage liver disease; TNM: tumor, node, metastasis.

**Table 7 jcm-14-03941-t007:** Categorization of influencing factors on HRQoL.

Category	Influencing Factors	Author and Year
Demographic factors	Age Gender Education Level Marital Status Employment Status Living Location Family Support	Chen et al., 2022 [11]; Shun et al., 2012 [22]; Zhao et al., 2024 [24]; Zheng et al., 2020 [25]; Liu et al., 2020 [26]
Clinical factors	TACE treatment Tumor type Tumor size Child-Pugh classification TNM stage History of hepatitis or liver resection	Hartrumpf et al., 2018 [20]; Chen et al., 2022 [11]; Zhao et al., 2024 [24]; Cao et al., 2012 [27]
Psychological factors	Anxiety Depression Sadness Spiritual health Coping mechanisms	Hartrumpf et al., 2018 [20]; Shun et al., 2012 [22]; Zhao et al., 2024 [24]; Cao et al., 2012 [27]
Social support factors	Family income Family support Social support	Zhao et al., 2024 [24]; Zheng et al., 2020 [25]
Physical factors	Fatigue Abdominal distension Pain Appetite loss	Chen et al., 2022 [11]; Cao et al., 2012 [27]

## Data Availability

No new data were created or analyzed in this study.

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
