# Peer review of "Determinants of Health-Related Quality of Life After Transarterial Chemoembolization in Hepatocellular Carcinoma Patients: A Systematic Review"

_jcm, 2025, doi:10.3390/jcm14113941_

Round 1

Reviewer 1 Report

Comments and Suggestions for Authors

This systematic review addresses an important and clinically relevant topic by synthesizing evidence on determinants of health-related quality of life (HRQoL) in hepatocellular carcinoma (HCC) patients undergoing transarterial chemoembolization (TACE). The study is well-structured, adheres to PRISMA guidelines, and employs a rigorous methodology. However, several areas require clarification or improvement to strengthen the manuscript’s scientific contribution and readability.

  1. Literature Search and Selection Criteria
    • While the inclusion of both English and Chinese databases (e.g., CNKI, Wanfang) enhances the scope of the review, the search strategy lacks justification for excluding non-observational studies (e.g., qualitative or mixed-methods research). A brief rationale for limiting the review to observational designs would improve transparency.
  2. Heterogeneity and Synthesis
    • The narrative synthesis effectively categorizes factors into five domains. However, the heterogeneity in HRQoL assessment tools (e.g., EORTC QLQ-C30, FACT-G, SF-12) and follow-up durations (3 days to 12 months) complicates cross-study comparisons. A discussion of how these variations might affect the validity of the synthesized findings is warranted.
    • Table 1 lists studies from 2012 to 2024. Given the rapid evolution of TACE techniques (e.g., drug-eluting beads, combination therapies), a brief discussion of whether technological advancements might influence HRQoL outcomes over time would contextualize the findings.
  3. Clinical Implications
    • The recommendation for “multidimensional interventions” is well-supported but lacks specificity. For example, how might spiritual support (identified as a protective factor) be operationalized in clinical practice? Including concrete examples or referencing existing intervention frameworks (e.g., palliative care models) would enhance translational relevance.
  4. Quality Assessment
    • The JBI tools are appropriately applied, but the description of quality assessment results is overly condensed. A more detailed summary of common methodological weaknesses (e.g., confounding adjustment, exposure measurement) in Tables 2 and 3 would help readers evaluate potential biases.

Author Response

Thank you for your comments. Our response is attached. 

Reviewer 2 Report

Comments and Suggestions for Authors

The aim of this systematic review was to identify and critically evaluate and discuss the available information regarding factors that influence health-related quality of life (HRQoL) in HCC patients post-TACE. The Authors have considered the following English and Chinese databases: PubMed, Scopus, Web of Science, CNKI, and Wanfang. They selected only observational studies and studies that examined factors affecting HRQoL in HCC
patients, following TACE treatment. Overall, 9 studies satisfied the inclusion criteria, including 6 cohort and 3 cross-sectional studies. Of the HRQoL assessment tools, the EORTC QLQ-C30 and  FACT-G were the most common. The factors influencing HRQoL were grouped into 5 categories, such as demographic factors, clinical indicators (liver function, tumor burden, psychological factors, social support, and physical symptoms (fatigue, pain, appetite loss). Across studies, both symptom severity and psychological distress were consistently associated with lower HRQoL. The authors concluded that the HRQoL of HCC patients following TACE is influenced by a complex interplay of demographic, clinical, psychological, social, and symptomatic factors, thus suggesting that multidimensional interventions addressing these diverse aspects are relevant to optimizing recovery and improving overall well-being.

The study is of interest and addresses a clinically important topic. However, some issues require additional information and should be addressed.

-According to literature data, optimal candidates for TACE treatment are those having a preserved (or minimally deteriorated) liver function to avoid that the transarterial treatment might cause a liver function deterioration, with worsening of the Child-Pugh score. For this reason, current guidelines recommend using a selective or superselective transarterial approach for TACE. However, the authors did not report information about such a clinically relevant point. Liver function deterioration after TACE might have a significant impact on the patient's QoL, and this point should be discussed.

-Discussing the predictors influencing HRQoL, the authors should further discuss the impact of abdominal pain as well as fever. Both abdominal pain and fever post-TACE are frequently related to the "post-embolization" syndrome, which is also characterized by a transient serum hypertransaminasemia. However, according to recent literature data, a transient post-TACE hypertransaminasemia is significantly correlated with objective radiological response, thus offering a simple prognostic tool for tailored patient management, as recently demonstrated (doi: 10.3390/jpm11101041). This is an important point to recall in the discussion.

Author Response

(The authors gave the same response as above.)
